# Differential LRRK2 Signalling and Gene Expression in WT-LRRK2 and G2019S-LRRK2 Mouse Microglia Treated with Zymosan and MLi2

**DOI:** 10.3390/cells13010053

**Published:** 2023-12-26

**Authors:** Iqra Nazish, Adamantios Mamais, Anna Mallach, Conceicao Bettencourt, Alice Kaganovich, Thomas Warner, John Hardy, Patrick A. Lewis, Jennifer Pocock, Mark R. Cookson, Rina Bandopadhyay

**Affiliations:** 1Reta Lila Weston Institute of Neurological Studies and Department of Movement neuroscience, UCL Queen Square Institute of Neurology, London WC1N 1PJ, UKt.warner@ucl.ac.uk (T.W.); 2Center for Translational Research in Neurodegenerative Disease, Department of Neurology, University of Florida, Gainesville, FL 32610, USA; adamantios.mamais@neurology.ufl.edu; 3Department of Neuroinflammation, UCL Queen Square Institute of Neurology, University College London, London WC1N 1PJ, UK; anna.mallach@ucl.ac.uk (A.M.); j.pocock@ucl.ac.uk (J.P.); 4Department of Neurodegenerative Diseases, UCL Queen Square Institute of Neurology, University College London, London WC1N 3BG, UK; c.bettencourt@ucl.ac.uk (C.B.); j.hardy@ucl.ac.uk (J.H.); plewis@rvc.ac.uk (P.A.L.); 5Cell Biology and Gene Expression Section, National Institute on Aging, Bethesda, MD 20892, USA; kaganovichal@mail.nih.gov (A.K.); cookson@mail.nih.gov (M.R.C.); 6Royal Veterinary College, University of London, London NW1 0TU, UK

**Keywords:** *LRRK2*, TLR2, LRRK2 p.G2019S knock-in, zymosan, MLi.2, RNA-Seq

## Abstract

Mutations in the leucine-rich repeat kinase 2 (*LRRK2*) gene cause autosomal dominant Parkinson’s disease (PD), with the most common causative mutation being the *LRRK2* p.G2019S within the kinase domain. LRRK2 protein is highly expressed in the human brain and also in the periphery, and high expression of dominant PD genes in immune cells suggests involvement of microglia and macrophages in inflammation related to PD. LRRK2 is known to respond to extracellular signalling including TLR4, resulting in alterations in gene expression, with the response to TLR2 signalling through zymosan being less known. Here, we investigated the effects of zymosan, a TLR2 agonist and the potent and specific LRRK2 kinase inhibitor MLi-2 on gene expression in microglia from *LRRK2-WT* and *LRRK2* p.G2019S knock-in mice by RNA-sequencing analysis. We observed both overlapping and distinct zymosan and MLi-2 mediated gene expression profiles in microglia. At least two candidate genome-wide association (GWAS) hits for PD, CathepsinB (*Ctsb*) and Glycoprotein-nmb (*Gpnmb*), were notably downregulated by zymosan treatment. Genes involved in inflammatory response and nervous system development were up and downregulated, respectively, with zymosan treatment, while MLi-2 treatment particularly exhibited upregulated genes for ion transmembrane transport regulation. Furthermore, we observed that the top twenty most significantly differentially expressed genes in *LRRK2* p.G2019S microglia show enriched biological processes in iron transport and response to oxidative stress. Overall, these results suggest that microglial LRRK2 may contribute to PD pathogenesis through altered inflammatory pathways. Our findings should encourage future investigations of these putative avenues in the context of PD pathogenesis.

## 1. Introduction

The etiopathology of PD is multifactorial, with a complex interplay of genes and environmental triggers [1,2]. Over the past decades, research has increasingly provided evidence that links neuroinflammation to neurodegeneration in PD. For example, microglia positive for human leukocyte antigen D (HLA-DR) have been identified in the substantia nigra pars compacta of PD patients [3]. Furthermore, there is an increase in inflammatory biomarkers including tumour necrosis factor alpha (TNFα), interleukin (IL)-1β, IL-6, epidermal growth factor (EGF), and transforming growth factor alpha (TGFα) in the brain and cerebrospinal fluid (CSF) of PD patients [4,5]. Similarly, increased IL-1β, IL-6, and gamma delta+ T cells, implicated in inflammation, occurs in the blood and brain of PD patients [6,7], supporting the potential involvement of immunological events in neurodegeneration. 

Mutations in the *LRRK2* gene cause autosomal dominant disease with clinical features that are indistinguishable from sporadic PD. The majority of the well-characterized pathogenic mutations in *LRRK2* are found in the two active catalytic domains (GTPase and kinase) of the encoded large multidomain protein. The common *LRRK2* p.G2019S mutation is located within the kinase domain and increases LRRK2 kinase activity [8]. LRRK2 protein is expressed within the brain and in peripheral tissues, with notably high expression in microglia and macrophages [9]. High expression of dominant PD genes within microglia and/or macrophages suggests that inflammation is relevant for PD development. Knockdown, knockout, or pharmacological inhibition of LRRK2 functions results in a decrease in inflammatory responses [10,11], further highlighting a potential role of LRRK2 in immune function [12]. Based on these observations, LRRK2 has been identified as a potential therapeutic target for anti-inflammation strategies in PD [13]. 

Inflammatory pathways are typically organized where cell surface receptors respond to external signals to trigger intracellular signalling events that result in alterations in gene expression. Prior work has shown that LRRK2 is responsive to extracellular signalling including TLR2 and TLR4 [14]. Activation of both TLR2 and TRL4 leads to marked phosphorylation of LRRK2 at Ser910 and Ser935 residues, resulting in recruitment of 14–3–3 proteins and re-localisation of LRRK2 in types of myeloid cells [15,16,17]. Zymosan, a known TLR2 agonist, increased LRRK2 localisation and phosphorylation at Ser910 and Ser935, potentially mediated via TLR2 signalling [16,18]. Here, we have investigated the effects of the TLR2 agonist zymosan on gene expression in microglia from *LRRK2* p.G2019S knock-in mice by RNA-Seq analysis. We demonstrate that zymosan triggered significant upregulation in LRRK2 phosphorylation, which remained significantly inhibited with MLi-2 treatment. We also show that the two genotypes, wild-type and G2019S-*LRRK2*, and two treatments, zymosan and MLi-2, induced overlapping as well as distinct effects on the gene expression profile of microglial cells. Zymosan consistently displayed a stronger effect on gene expression as compared to MLi-2 treatment and the G2019S-LRRK2 genotype. 

## 2. Materials and Methods

### 2.1. Animals and Primary Microglia Cultures and Treatment

Homozygous *LRRK2* G2019S knock-in mice [19,20] and C67Bl/6J wild-type mice were housed at NIH, and all animal procedures were carried out in strict accordance with the guidelines provided by the Care and Use of Laboratory Animals of NIH, as approved by the Institutional Animal Care and Use Committees of the US National Institute on Aging (Approval number: 463-LNG-2018). 

Primary mixed glial cultures were established from the brains of wild-type or from knock-in mice harbouring the G2019S-*LRRK2* mutation at postnatal days 1–2 (P1–2). After approximately 10 days, primary microglia were shaken from mixed glial cultures, as described in Russo et al. [19]. Microglia was isolated from 4–6 pups for each independent primary culture per genotype for biochemistry or RNA extraction. The mouse breeding schedule for WT and G2019S LRRK2 mice was matched to the day of microglia isolation through cohorts. The purity of the obtained culture was verified using the double immunofluorescence method by staining the culture with rabbit anti-Iba1 (Abcam, Cambridge, UK, #ab5076) for microglia, rabbit anti-glial fibrillary acidic protein (GFAP) for astrocytes (Dako Omnis, Santa Clara, USA,#GA524), and anti- 4′,6-diamidino-2-phenylindole (DAPI) (Cell Signaling, Danfoss, MA, USA, #4083). The primary microglial yield was approximately 5 × 10^5^ cells/flask with minimal astrocyte contamination. For immunoblotting, microglia were seeded and treated with pro-inflammatory agents for 4 h and 24 h. For RNA-Seq, cells were seeded and treated with pro-inflammatory agents for 24 h only. For immunoblotting, cells were lysed with the lysis buffer consisting of 10% cell lysis buffer (10×) (cell signalling #9803S), 1% protease inhibitor cocktail (100×) (cell signalling #5871S) and 1% phosphatase inhibitor cocktail (100×) (cell signalling #5870S), centrifuged at 3200× *g* for 5 min, collected and stored at −80 °C for future use. For RNA-Seq collection, RNA was extracted from the primary microglia using TRIzol reagent (Invitrogen, Waltham, MA, USA, #15596026), per the manufacturer’s protocol.

### 2.2. Protein Assay

Protein levels were measured using a Thermofisher Pierce BSA Standard Pre-Diluted Set, per the manufacturer’s instructions, using BSA as standard.

### 2.3. Zymosan and MLi-2 Treatment Protocol

Zymosan A from *Saccharomyces cerevisiae* (Sigma, St. Louis, MO, USA, #Z4250) and MLi-2 from Merck (Tocris, Bristol, UK, #5756) were used for these experiments as TLR2 inflammatory stimuli and an LRRK2 kinase inhibitor, respectively. For immunoblots, primary microglia from wild-type and G2019S-LRRK2 Tg mice were treated with 200 µg/mL zymosan and 1 µM MLi-2 for 4 h and 24 h, and for RNA-Seq, they were treated with the same concentration of zymosan and MLi-2, but for 24 h only. Zymosan dose was chosen based on previous studies that have used 200 μg/mL concentration of zymosan in immune cells, such as in BMDMs [16,20]. For MLi-2, previous studies have used a 1 mM concentration of MLi-2 in vivo [21] and a 0.5 mM concentration of MLi-2 in vitro [22].

### 2.4. Immunoblots and Fluorescent Blots

Samples were heated to 95 °C for 3 min in loading buffer (NuPAGE LDS Sample Buffer) (4×) (Invitrogen, NP0008) with 5% β-Mercaptoethanol, then loaded on 4–20% Criterion TGX Precast Midi protein gels (Bio-Rad, Hercules, CA, USA #5671094). The gels were run in TGS buffer (1×) and transferred using Trans-Blot Turbo Mini 0.2 µm Nitrocellulose Transfer Packs (Bio-Rad #1704158) using the Trans-Blot Turbo Transfer System (Bio-Rad). Membranes were incubated with primary antibodies overnight and β-actin for 1 h. Membranes to be incubated with LRRK2 and Ser935 were run on separate blots with their respective β-actin. After the membranes were incubated with the appropriate fluorescent-conjugated secondary antibodies, protein bands were visualised using Odyssey CLx (Li-Cor) and quantified using Bio-Rad Lab Image (version 6.1) software.

### 2.5. RNA Sequencing (RNA-Seq)

RNA quality and integrity were measured using an Agilent 2100 Bioanalyzer RNA 6000 Nano Chip (Agilent, Santa Clara, CA, USA) and analysed on Agilent 2100 Expert Software 228259. The samples were then stored at −80 °C until further use. The RNA-Seq was carried out by Psomagen labs, whole-genome sequencing service providers in the USA, through Illumina SBS technology using a Total RNA Ribo-Zero Gold Library Kit.

### 2.6. Analysing RNA-Seq Data

RNA-Seq reads were aligned to the mouse reference genome (mm10) using STAR [23], and expression counts per transcript were quantified using eXpress [24] followed by using DESEQ2 [25] to normalise data. Gene expression data were loaded into MATLAB version R2019b to identify upregulated and downregulated genes in different treatment and genotype groups through differential gene expression analysis. Subsequently, the functional enrichment analysis tools FunRich version 3.1.3 and Hippie version 2.2 were used to perform functional enrichment and interaction network analysis on the identified upregulated and downregulated genes in different treatment groups, and to map genes.

### 2.7. Statistical Analyses

A one-way analysis of variance (ANOVA) with Tukey’s post hoc test was used to compare the difference between the means of three or more treated and untreated independent groups. Unpaired two-tailed student’s t-tests were used to compare the difference between the means of any two normally distributed groups, e.g., when determining the difference between non-stimulated cells and cells stimulated with zymosan. The statistical tests used for each data set are mentioned where appropriate in the legends of every experimental figure. All data were analysed using GraphPad Prism 5, R Studio version 1.3.1093, and MATLAB version R2019b. All quantitative data are expressed as mean ± S.E.M, and represent in most cases at least three independent sets of experiments. The exact number of experiments with internal replicates is mentioned in the legends of every experimental figure. Statistical significance was set at *p* < 0.05.

## 3. Results

### 3.1. Zymosan Induces Phosphorylation of LRRK2 in Microglia

LRRK2 is phosphorylated in a series of residues between the ankyrin and leucine-rich repeat domains, including Serine935 [26]. Phosphorylation of LRRK2 at S935 is controlled by other kinases, notably casein kinase 1α [27], but is also responsive to pharmacological inhibition of LRRK2 kinase inhibitors [15,16]. We therefore used S935 phosphorylation as an indirect measure of LRRK2 kinase activity. A significant upregulation of LRRK2 phosphorylation at Ser935 was observed with zymosan treatment at 4 h and 24 h (Figure 1) in wild-type or G2019S knock-in cells. Co-treatment of cells with zymosan and MLi-2 resulted in a significant decrease in pS935-LRRK2 levels at 4 h in both genotypes, and at 24 h in wild-type cells.

### 3.2. Differential Gene Expression between Wild-Type and G2019S-LRRK2 with Zymosan and MLi-2

Having established that zymosan treatment results in an increase in pS935 LRRK2, we next performed RNA-Seq to evaluate the effects of LRRK2 modulation on inflammatory signalling in two genotype groups (WT and G2019S) and four treatments (control, MLi-2, zymosan and zymosan plus MLi-2). Mean vs. variance plots for gene expression after variance stabilizing transformation indicate that variance was consistent across gene expression levels in both genotypes (Appendix A).

We next examined the first two principal components of the overall gene expression patterns against genotype and treatment groups. PC1, which explains ~90% of the variance, aligns with zymosan treatment, with or without the addition of MLi-2, while PC2, which captures ~5% of the variance, appears to be largely explained by genotype (Figure 2). Consistent with this overall view of the data, clustering of overall gene expression by Euclidean distance shows a clear separation between zymosan-treated samples and samples without zymosan, followed by separation by genotype, and finally MLi-2 treatment (Figure 3A). Examination of the top twenty differentially expressed genes also shows separation between treatments (Figure 3B). 

We next examined gene expression between treatment groups. Volcano plots show significantly differentially expressed genes with zymosan treatments in both wild-type and G2019S-LRRK2 genotypes (Figure 4A,B). There were abundant differences induced by zymosan treatment, including *Fth1* and lysosomal genes, which we have previously shown to be responsive to LPS-induced inflammation in microglia [28]. We also note that at least two candidate GWAS hits for PD, namely *Ctsb*) and *Gpnmb*, are downregulated by Zymosan treatment, irrespective of genotypes. Protein interaction maps of Gpnmb and Ctsb are shown in Figure 5.

### 3.3. Functional Enrichment Analysis in Wild-Type and G2019S-LRRK2 with Zymosan and MLi-2

Two protein interaction maps generated with Hippie software show the interaction of *Ctsb* and *Gpnmb* proteins with other interacting proteins. Ctsb and Gpnmb from the list of top twenty genes (Figure 3A,B) are also GWAS hits for PD [29]; thus, it is interesting to see their interacting proteins in the context of PD.

Ontological analysis of significant differentially expressed genes was performed using FunRich software (version 3.1.3). Figure 6 shows biological processes (A,B) and cellular components (C,D) significantly enriched in zymosan-treated wild-type and G2019S-LRRK2 microglia. The most enriched GO term for biological processes in upregulated genes was inflammatory response, and in downregulated genes, it was nervous system development. The most enriched GO term for cellular components was plasma membrane for upregulated genes, and glutamatergic synapse for downregulated genes.

Furthermore, Figure 7 shows the biological processes and cellular components significantly enriched in MLi-2 treated wild-type and G2019S-LRRK2 microglia. The most enriched GO term for biological processes in upregulated genes was regulation of ion transmembrane transport, and in downregulated genes, this was positive regulation of apoptotic processes. The most enriched GO terms for cellular components were plasma membrane for downregulated genes, and no significantly enriched cellular components terms for upregulated genes (Appendix A). Notably, Figure 8 shows (A) biological processes and (B) cellular components significantly enriched for the top twenty most significantly differentially expressed genes in G2019S-LRRK2 microglia. The most enriched GO terms for biological processes in upregulated genes were iron ion transport and response to oxidative stress; the most enriched GO term for cellular components was extracellular space.

### 3.4. Basal Level of Significant Differentially Expressed Genes in the Wild-Type and G2019S-LRRK2 Genotype

We also studied the basal-level gene expression in wild-type and G2019S-LRRK2 genotype, and found that 362 genes were upregulated in the G2019S-LRRK2 genotype as compared to 1044 genes in wild-type cells, with 32 genes upregulated in both. On the other hand, 251 genes were downregulated in G2019S-LRRK2 genotype as compared to 1725 genes in wild-type with eight genes downregulated in both. Results are displayed as Venn diagrams in Appendix A. We also performed the analysis for the genes present in G2019S, but not in wild-type LRRK2 levels. The results are displayed in Appendix A.

## 4. Discussion

Microglia are resident innate immune cells of the CNS, contributing neuroprotective characteristics during acute immune responses, but are also implicated as mediators of cell loss in neurodegenerative disorders after chronic activation. Activated microglia have been found in SNpc of PD patients [30], and are also implicated in causing damage to dopaminergic neurons [31]. The active role of microglia in neuroinflammation is also reviewed in a study by Perry et al. [32]. Upon activation by inflammatory stimuli, microglia switch from resting to activated state, releasing pro-inflammatory and reactive oxygen species to mediate an inflammatory response [33]. LRRK2 is highly expressed in microglia [9], indicating a possible role for LRRK2 and microglia in contributing to PD pathogenesis through altered inflammatory signalling. The inflammatory stimulus LPS is widely used to trigger microglial activation, as previously carried out [14], and is used to analyse the transcriptomic profile of primary microglial cells by various studies [34,35]. To our knowledge, transcriptomic data using zymosan as an inflammatory stimulus in the context of LRRK2 in mouse primary microglia have not been investigated to date.

To dissect novel LRRK2-related biological processes in microglia, we performed RNA-Seq analysis of wild-type and G2019S-LRRK2 cells under basal conditions, or after stimulation with zymosan and in the presence of the LRRK2 kinase inhibitor, MLi-2. We observed various overlapping and individual effects on gene expression and induction of biological processes. Our data have shown zymosan to have a stronger effect on gene expression as compared to MLi-2 treatment or genotype. This was confirmed with first principle component of the overall gene expression profile, which separated zymosan from MLi-2 and basal treatment groups, whereas genotype had a more subtle effect on overall gene expression. This was similar to a previous transcriptomic study using LPS and α-synuclein PFFs [34], where genotype had a subtler effect on overall gene expression. The results are unsurprising, given that zymosan is a strong stimulator of immune signalling similar to LPS, with LRRK2 likely playing a modulatory role through altered lysosomal responses [28]. Also similar to the same study, the first principle component of the overall gene expression profile in our study separated zymosan from the other two treatment groups (MLi-2 and control), while the second principle component separated the genotypes. These findings show that both zymosan and genotype induce gene expression responses, but that they differ from each other.

Zymosan particles are recognized by a variety of receptors on macrophages including TLR2 and 6, mannose receptor, dectin 1, and complement receptor 3 [36,37]. Consequently, phagocytosis of zymosan is a complex sequence of events that involves various signalling cascades. Phagocytosis of zymosan is accompanied by actin reorganization, which drives the extension of pseudopodia (phagocytic cups) around zymosan particles. It has been shown that the regulation of actin reorganization during phagocytosis of zymosan depends on Rho-family GTPases, such as Rac1 and Cdc42 [38,39]. Further studies have shown that zymosan affects various intermediates in the c-AMP signalling pathway. In macrophage cells, cAMP inhibits the activation of NF-kB pathway induced by TLR stimuli, and reduces the production of pro-inflammatory cytokines [40]. Another report showed that zymosan is capable of stimulating the production of protein kinase A in bone-marrow-derived macrophages [41].

In the context of LRRK2 function in immune cells, zymosan has been shown to elevate LRRK2 phosphorylation at sites 910 and 935 in BMDMs, and this effect is independent of dectin 1[16]. In human induced pluripotent stem cells, LRRK2 operates at the intersection between phagosome maturation and recycling pathways in the myeloid lineage [18]. Diminished clearance of the bacterium *S. typhimurinum* was observed in vitro, and siRNA reduced LRRK2 levels in RAW macrophage cells [42]. Similarly, in LRRK2 KO mice, *S. typhimurinum* clearing was compromised [43]. This suggests that the presence of LRRK2 is necessary for a proper immune response. Intriguingly, the *G2019S* mutation conferred improved response to the infection [44].

Enrichment analyses showed that some of the most enriched GO terms for biological processes in zymosan-treated wild-type microglial cells were inflammatory response, innate immune response, and cytokine-mediated signalling pathways (Figure 4A). This was similar to gene ontology analysis in the transcriptomic study involving LPS where a number of responses were related to inflammation, as expected [34]. However, these GO terms were not significantly enriched in G2019S genotype, which is suggestive of the potential involvement of G2019S-LRRK2 (hyperkinetic) in inflammatory responses. Likewise, in the same study, a category involving regulation of reactive oxygen species’ metabolic processes was enriched in PFFs-treated cells, but not LPS-treated cells, which was also not enriched in zymosan-treated cells in our study. This indicates that both zymosan and LPS induce similar and specific effects on gene expression, possibly by recruiting receptors with similar pathways in response to the two inflammatory insults (LPS and zymosan). It will also be interesting to study the effect of zymosan in *Lrrk2*-deficient microglia, since LPS was studied in *Lrrk2*-deficient microglia in another transcriptomic study to dissect any similarities or differences between LPS and zymosan and LRRK2 functions. Pathways significantly enriched with zymosan treatment in the G2019S genotype involved the adrenomedullin receptor signalling pathway, the cyclic guanosine monophosphate (cGMP) biosynthetic process, the receptor guanylyl cyclase signalling pathway, and the neuropeptide signalling pathway, which were not observed in wild-type cells (Figure 4A). Therefore, it may be important to investigate these pathways in the context of PD pathogenesis. Adrenomedullin recently been discussed as an important participant in neurological diseases, and has been seen to exhibit a neuroprotective effect against brain insults [45]. Similarly, guanylyl cyclase–cGMP signalling and the neuropeptides and neurotransmitters are also known to play a role in PD pathogenesis [46,47]. GO-enriched terms also indicate zymosan to be a positive regulator of ERK1 and ERK2 signalling cascade and apoptotic processes. ERK kinases belong to the MAPK superfamily of kinases, which are known to be implicated in PD [48]. The most significantly enriched terms for biological processes for downregulated proteins with zymosan treatment in G2019S genotype cells include nervous system development, regulation of ion transmembrane transport, and modulation of chemical synaptic transmission, indicating these processes to perhaps be negatively affected in PD.

In terms of cellular processes, the most enriched terms for zymosan treatment the in wild-type and G2019S genotype include plasma membrane, extracellular space, and glutamatergic synapse. MLi-2, on the other hand, seemed to have a more subtle effect on expressed genes and enriched GO terms. Interestingly, for enriched biological processes in G2019S, upregulated genes in MLi-2 seemed to only significantly enrich processes involved in myelination. However, the underlying mechanisms of the inflammatory effects and role of microglia are yet to be fully understood in the context of PD.

Furthermore, in the basal G2019S condition, downregulated proteins were significantly enriched for chemotactic pathways such as natural killer cells, lymphocytes, and eosinophil chemotaxis. This is suggestive of suppressed chemotactic abilities in PD pathogenesis, and indeed, a recent study has found lymphocyte chemotaxis to be impaired in PD patients [49]. Furthermore, it also suggests that microglia expressing G2019S have an inability to become recruited to areas of damage within the PD brain parenchyma. A similar microglial response is linked to the AD risk gene TREM2 [50]. Defence response to viruses was also significantly enriched for downregulated proteins in the G2019S genotype, which confirmed impaired response to viruses as one of the key contributors to PD pathogenesis. Previous studies have shown that systemic and local inflammatory responses to viruses have potential involvement in neuronal damage, even in the absence of neuronal cell death. Viruses are seen to evoke CNS inflammation either by entering the brain, or by crossing a damaged BBB or along the peripheral nerves, or by activating peripheral innate and adaptive immune responses [51,52]. Cellular components most significantly enriched in upregulated proteins in the G2019S genotype also consist of the plasma membrane, myelin sheath, and internode and paranode regions of axons, rendering investigating all these biological processes and cellular components important in the context of PD [53,54].

The top twenty genes which showed the highest mean expression across all genotype and treatment conditions also revealed some intriguing aspects of the results. First, two of the twenty genes, *GPNMB* and *CTSB*, were reported as genetic risk loci for PD in a recent GWAS study [29]. Elevated expression of *GPNMB* is associated with increased PD risk [55] and through its interaction with alpha-synuclein, it may also confer increased disease risk [56]. GPNMB protein is expressed in microglia, and it is elevated in the brains of Alzheimer’s patients, although the relevance of this in neurodegeneration remains ambiguous [57]. *CTSB* is a risk factor in PD GWAS, and *CTSB* variants have been found to affect susceptibility to PD, albeit with differing penetrance [58]. Additionally, risk variants in the *CTSB* locus were identified to decrease its mRNA expression [59]. CTSB is a lysosomal cysteine protease that play critical roles in pathophysiological processes of protein degradation, lysosomal biology, energy metabolism, and immune response [60,61]. CTSB also indirectly controls TFEB, an important regulator of autophagy and lysosomal gene expression [60]. Moreover, cysteine cathepsins are essential in degradation of alpha-synuclein within lysosomes [62]. However, CTSB activity is context-dependent, and may be variable in different cell types [58]. Therefore, lower CTSB levels may contribute to the pathogenic pathways of PD. Further investigations are necessary to uncover the roles of GPNMB and CTSB in the pathophysiology of PD.

It is noteworthy that the Cathepsin D (*ctsd*) gene is mutated in human neuronal ceroid lipofuscinosis [63,64]. Interestingly, LRRK2 *G2019S* leads to suppression of lysosomal proteolytic activity in macropahges, and it regulates the abundance of multiple lysosomal proteins [65]. Therefore, research on Ctsb, Ctsd and Gpnmb in relation to LRRK2 and their interacting proteins should be expanded in order to further elucidate the molecular underpinnings of PD. Enriched biological processes for the top twenty genes show the most significantly enriched terms to be iron ion transport and response to oxidative stress, and enriched cellular components show extracellular space and lysosomes to be most significantly enriched (Figure 6). These are indeed important biological processes and cellular components implicated in PD.

## 5. Conclusions

With these data, our study is the first to report the identification of differential gene expression and dissection of novel biological pathways in response to zymosan and LRRK2 kinase inhibition with MLi-2 in G2019S-LRRK2 KI primary microglia cells. However, we acknowledge the important caveat of interpretation with data from an *n* of two experiments, which was dictated to some extent by a lack of resources (primary microglia). Nonetheless, the data presented in this paper raise several interesting points and open up new pathways of PD research that would be suitable for further investigation. In this study, we assessed LRRK2 activation via phosphorylation of Ser935—an indirect measure mediated by an upstream kinase. In future work, it will be important to assess direct measures of LRRK2 kinase activity such as phosphorylation of LRRK2 Ser1292, a site that is directly phosphorylated by LRRK2 kinase, but this could be challenging unless further steps are incorporated to visualize it using an immunoblot analysis [21], and phosphorylation of Rab proteins such as Rab10, Rab12 or Rab8a. One interesting future research direction could be to utilize human-derived cells with native LRRK2 expression. It would be necessary to understand whether human primary cells have the same phenotype as their mouse equivalents, or whether they diverge in phenotype. With the increasing use of stem cell-derived human cells to model disease, assessing zymosan-induced changes in expression in pluripotent stem cell-derived macrophages and microglia with and without G2019S-LRRK2 mutations will be insightful for a deeper, and more human-relevant, analysis. As noted above, the various enriched biological pathways discussed in this paper should be highlighted and prioritised for investigation in the context of PD pathogenesis—especially the regulation of genes implicated by genome-wide association studies of PD that are linked to lysosomal function. Finally, it is hoped that the findings from the research conducted for this paper will help advance the understanding of PD pathogenesis and drug development towards with the aim of therapies for the millions of people worldwide living with PD [66].

## Figures and Tables

**Figure 1 cells-13-00053-f001:**
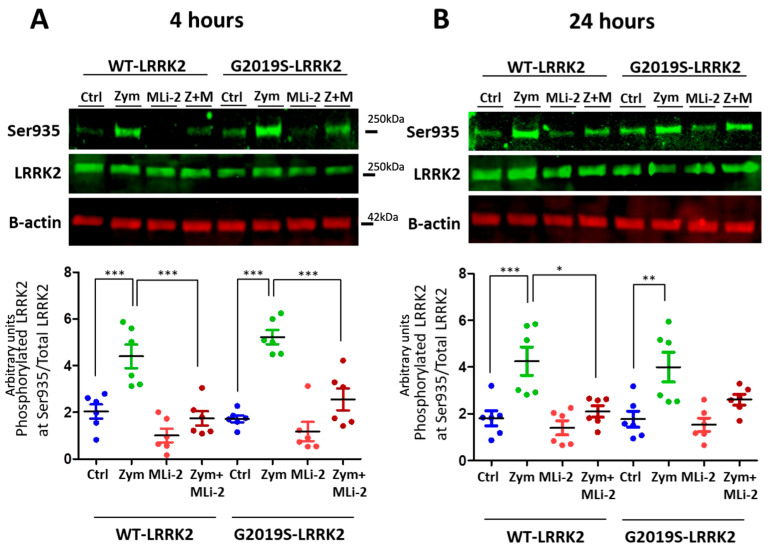
LRRK2 phosphorylation evoked by zymosan is inhibited with MLi-2 treatment in wild-type and G2019S-LRRK2 microglia. Wild-type and G2019S-LRRK2 were treated with 1 µM MLi-2 and 200 µg/mL zymosan for 4 h and 24 h. Fluorescent immunoblots and corresponding quantifications are shown for (**A**) 4 h and (**B**) 24 h. Blots were probed with LRRK2 phosphorylation antibody pSer935 and total LRRK2. Controls contain media only. Values represent the mean ± S.E.M. of two independent experiments (with internal triplicates in each experiment). The individual replicates are biologically independent, resulting in a sample size of *n* = 6. Data were analysed with a one-way ANOVA with Tukey’s post hoc test. * and ** and *** signify *p* < 0.05, 0.01 and 0.001, respectively. Colours for graphs indicate the following: blue: control; green: zymosan; red: MLi-2; dark red: zymosan + MLi2 treatments. Ctrl: control; Zym/Z: zymosan; M: MLi-2.

**Figure 2 cells-13-00053-f002:**
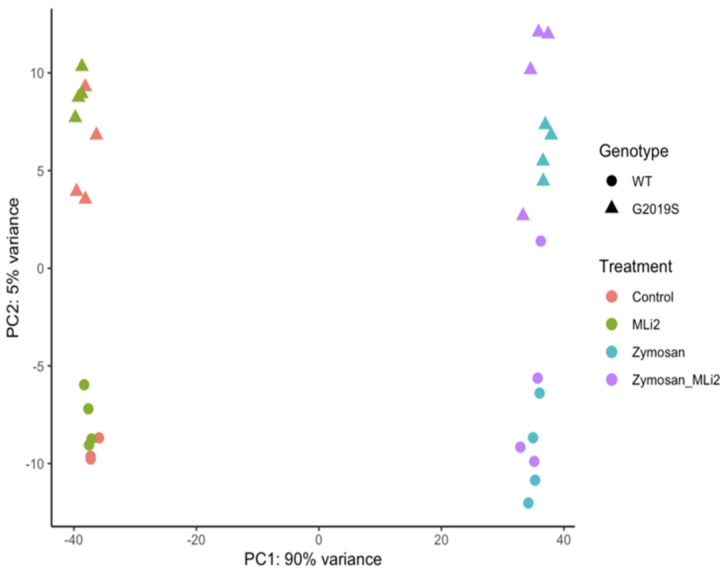
RNA-Seq profiling showing the first two principal components of wild-type and G2019S-LRRK2 microglia after zymosan and MLi-2 stimulation. We used the first two principal components, PC1 and PC2, of all samples in the study. Data are from two independent experiments. Treatments with zymosan and MLi-2 and genotype (wild-type and G2019S-LRRK2). Note that samples are separated largely by treatment, and to a lesser extent by genotype.

**Figure 3 cells-13-00053-f003:**
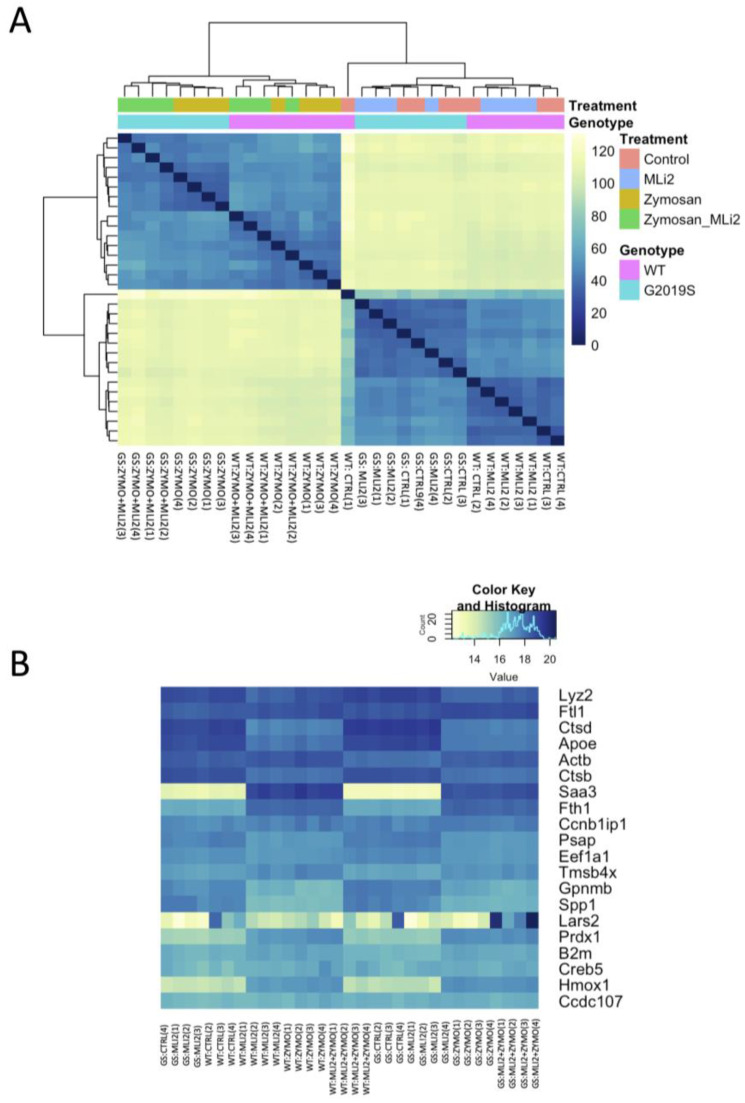
Differential gene expression in wild-type and G2019S-LRRK2 microglia after zymosan (200 μm) and MLi-2 (1 μm) stimulation for 24 h, (**A**) Hierarchical clustering and heatmap for the variance-stabilized expression of all genes detected in cells. Colours in the heatmap represent the Euclidean distance between samples in a pairwise manner, scaled as shown on the upper right yellow-blue scale. At the side of the heatmap is a colour representation of the model variables, which included two biological variables, and treatments with zymosan and MLi-2 and genotype (wild-type and G2019S-LRRK2) in two independent experiments with two biological replicates. Note that samples are separated largely by treatment and to a lesser extent by genotype. (**B**) Heatmap for the top twenty most statistically significant genes as examples of differential expression. Each gene on the right side of each heatmap is coloured according to Z (normalized standard deviations from the mean) for expression relative to the overall mean expression for that gene, and samples are listed below each heatmap. Ctrl: control; Zymo: zymosan.

**Figure 4 cells-13-00053-f004:**
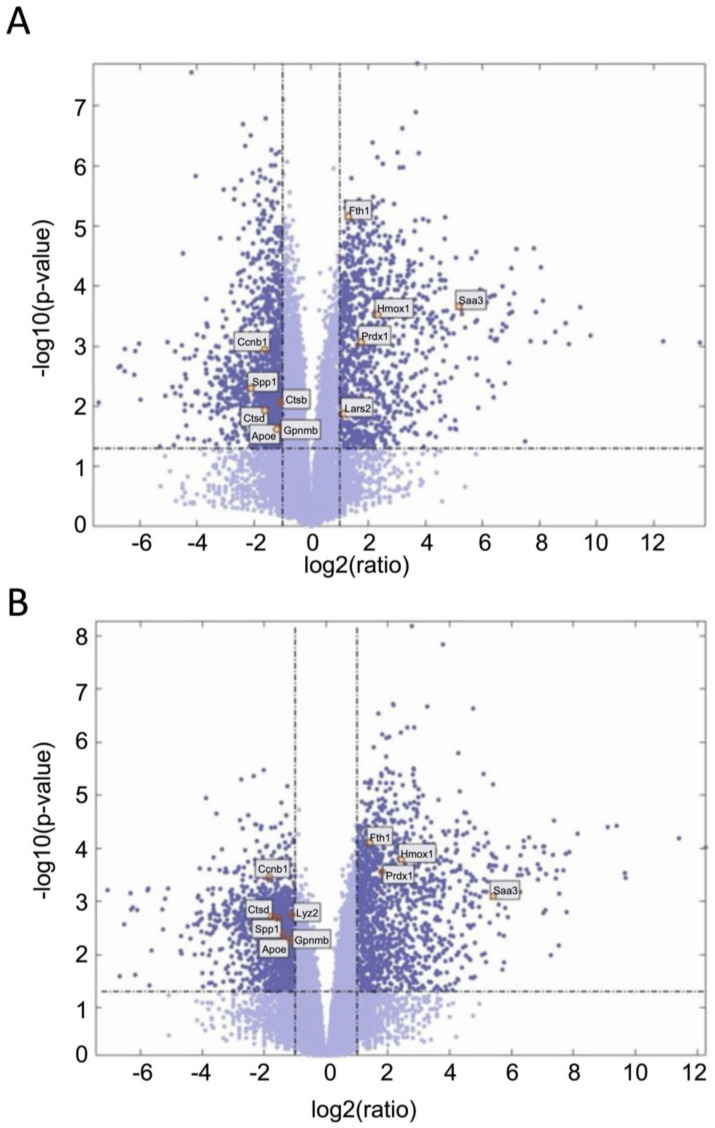
Volcano plots showing significantly differentiated genes with zymosan treatment in G2019S-LRRK2 (**A**) and in the wild type (**B**). Each point represents a significantly differentiated gene. Dark blue colour depicts genes which passed the thresholds for 2-Log fold change, with the upper-right-hand quadrant showing upregulated genes and the upper-left-hand quadrant showing downregulated genes. The top twenty genes with the highest mean expression across all samples are shown in boxes in these volcano plots.

**Figure 5 cells-13-00053-f005:**
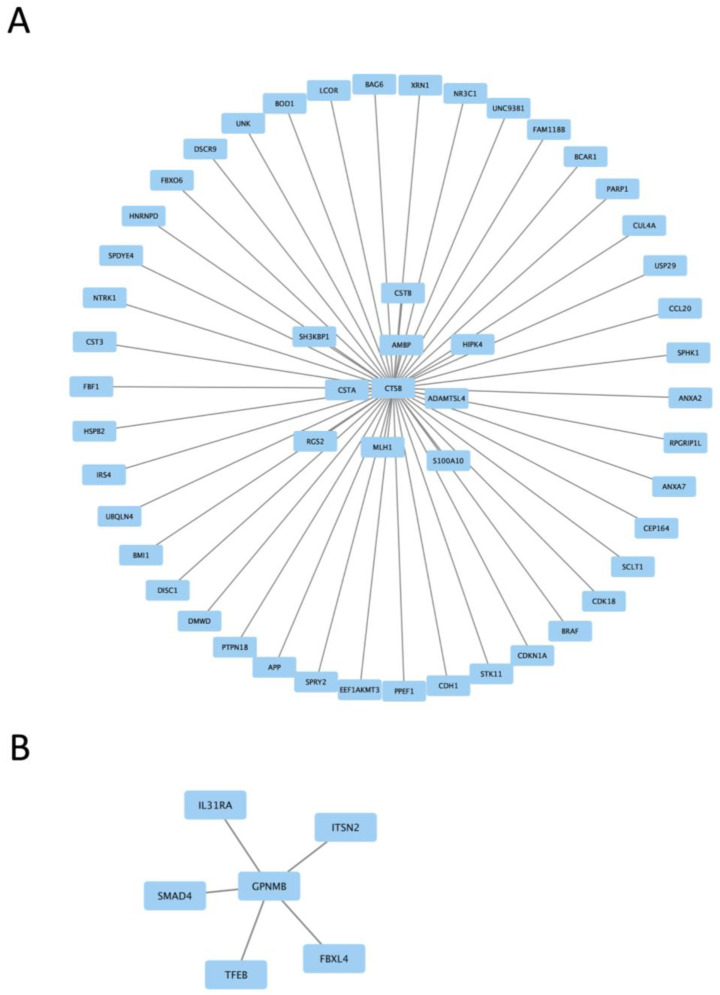
Protein interaction maps generated with Hippie software (version 2.2), showing the interaction of Ctsb proteins (**A**) and Gpnmb proteins (**B**) with other interacting proteins.

**Figure 6 cells-13-00053-f006:**
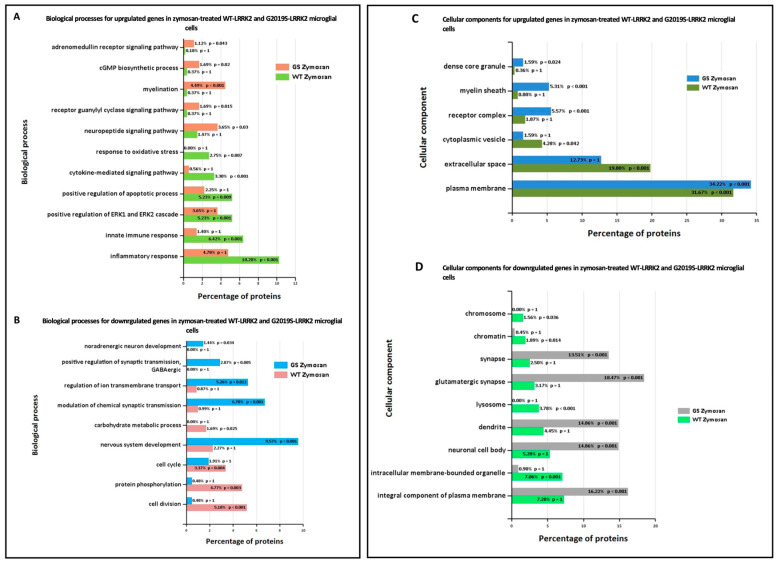
GO enrichment analysis of differentially expressed genes in zymosan-treated wild-type and G2019S-LRRK2 microglia. GO (**A**,**B**) biological process and (**C**,**D**) cellular component enrichment analyses of (**A**,**C**) upregulated proteins and (**B**,**D**) downregulated proteins in zymosan-treated wild-type and G2019S-LRRK2 microglial cells were performed using the FunRich functional enrichment analysis tool. Significantly enriched GO terms are shown with Benjamini–Hochberg and Bonferroni-corrected *p*-values. Statistical significance was taken at *p* < 0.05.

**Figure 7 cells-13-00053-f007:**
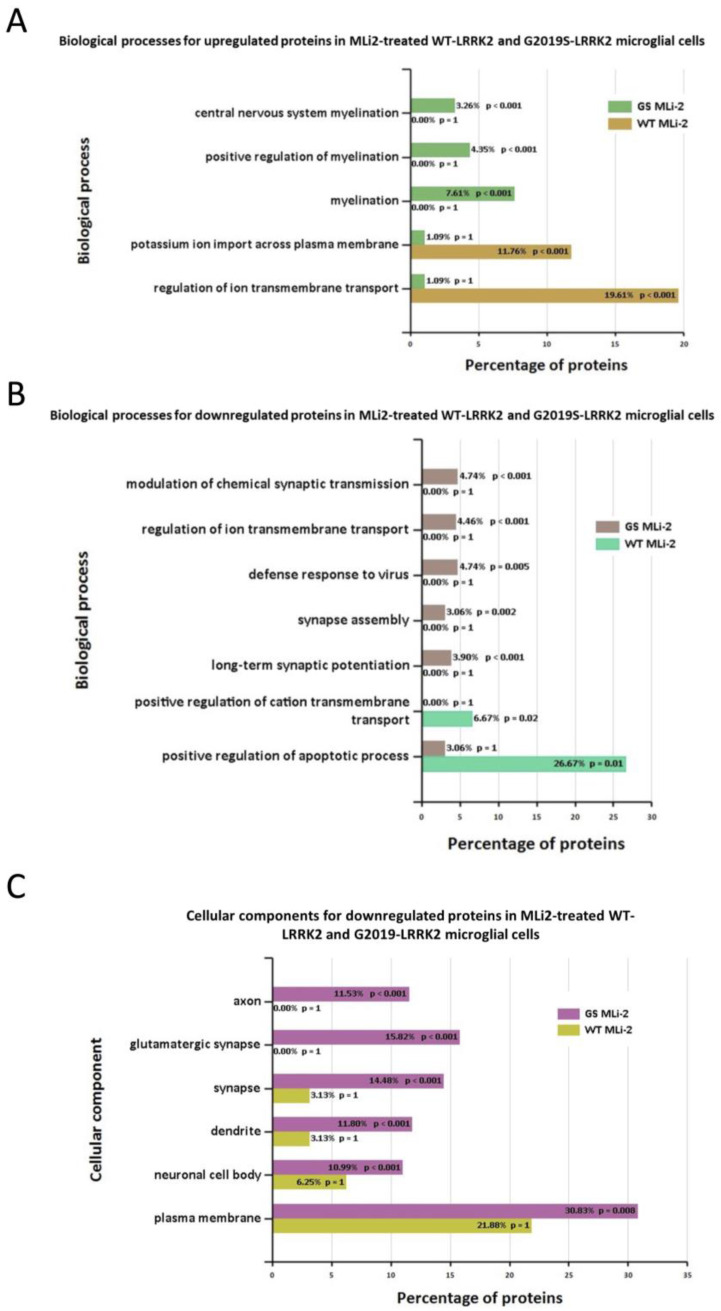
GO enrichment analysis of differentially expressed genes in MLi2-treated wild-type and G2019S-LRRK2 microglia. GO (**A**,**B**) biological process and (**C**) cellular component enrichment analyses of (**A**) upregulated proteins and (**B**,**C**) downregulated proteins in MLi2-treated wild-type and G2019S-LRRK2 microglial cells were performed using the FunRich functional enrichment analysis tool. There were no significantly enriched GO terms with upregulated proteins in MLi2-treated wild-type and G2019S-LRRK2 microglial cells in the data, hence no data for that are shown. Significantly enriched GO terms are shown with Benjamini–Hochberg and Bonferroni-corrected *p*-values. Statistical significance was taken at *p* < 0.05.

**Figure 8 cells-13-00053-f008:**
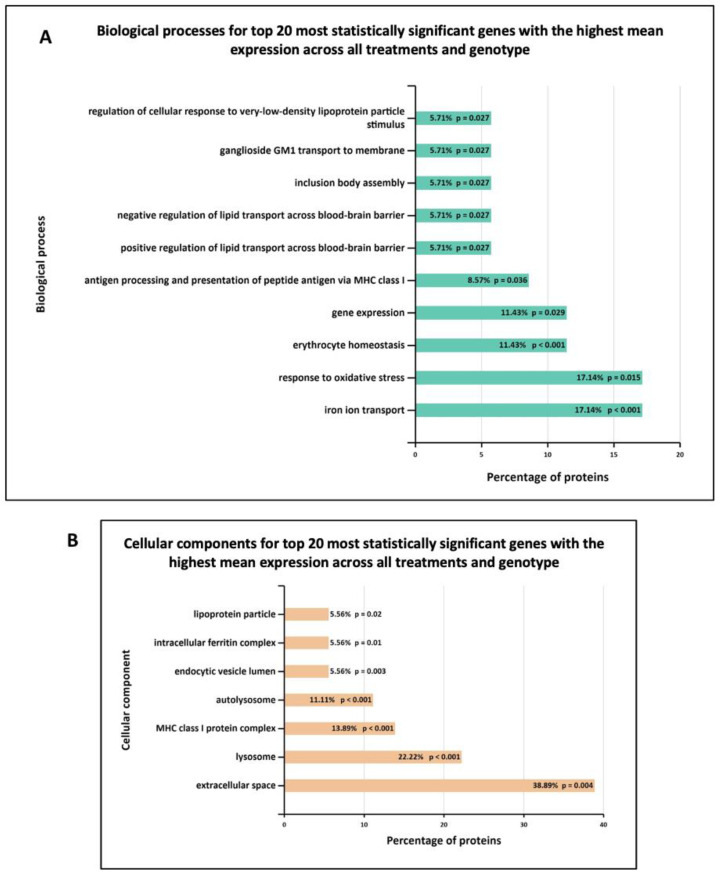
GO enrichment analysis of the top twenty most statistically significant genes with the highest mean expression across all treatments and genotype. GO (**A**) biological process and (**B**) cellular component enrichment analyses of top twenty most statistically significant differentially expressed genes across all samples (from Figure 5) were performed using the FunRich functional enrichment analysis tool. Significantly enriched GO terms are shown with Benjamini–Hochberg and Bonferroni-corrected *p*-values. Statistical significance was taken at *p* < 0.05.

## Data Availability

Data are contained within the article and Appendix A.

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
