# Peer review of "Differential LRRK2 Signalling and Gene Expression in WT-LRRK2 and G2019S-LRRK2 Mouse Microglia Treated with Zymosan and MLi2"

_cells, 2023, doi:10.3390/cells13010053_

Round 1
Reviewer 1 Report
Comments and Suggestions for Authors
Nazish et al., treat primary mouse microglia from WT and Lrrk2 p.G2019S mice with zymosan and MLi-2 to subsequently evaluate gene expression changes.
The authors show increased Lrrk2 activation, assessed by phosphorylation after zymosan treatment that is normalized with MLi-2. However, the authors do not evaluate the phosphorylation levels of any Lrrk2 substrate, that would be a good readout to ensure that Lrrk2 kinase activity is increased. Further, I think that the experiments should be performed independently a minimum of 3 times (not just 2 times), and show the results as the average of the technical replicates as just one single value.
Authors detect that the strongest transcriptonional perturbation is driven by treatment, irresepective of genotype. The authors should evaluate if there are basal gene expression differences between WT and Lrr2-G2019S microglia, or at least state that there are not, if that’s the case.
Also, the authors cuold integrate this dataset with the LPS treated microglia study.
THe interpretation of the chagnes in lysosomal genes is not clear. zymosan activate microglia, but downregulates Ctsb expression. Authors should discuss how this data agrees or disagrees with previous reports showing that increased expression of Ctsb is associated with increased PD risk.
Overall, rationale of the study is not clear. The evaluation of the consequences of zymosan-induced transcriptomic alterations in WT microglia and Lrrk2-G2019S microglia lacks biological significance.
Author Response
Nazish et al., treat primary mouse microglia from WT and Lrrk2 p.G2019S mice with zymosan and MLi-2 to subsequently evaluate gene expression changes.
The authors show increased Lrrk2 activation, assessed by phosphorylation after zymosan treatment that is normalized with MLi-2. However, the authors do not evaluate the phosphorylation levels of any Lrrk2 substrate, that would be a good readout to ensure that Lrrk2 kinase activity is increased. Further, I think that the experiments should be performed independently a minimum of 3 times (not just 2 times), and show the results as the average of the technical replicates as just one single value.
Response: We thank the reviewer for his insightful comments. As we used primary microglia from mice, the yields were quite small to use in three independent experiments for immunoblot. For the same reason, we could not use immnoblot analysis for bonafide LRRK2 kinase substrates such as RABs or LRRK2-PSER1292. We can clarify that we have performed 2 independent experiments with 3 or 2 biological replicates as detailed in the legends of each figure and therefore the statistics and the representative graphs described is justified. These are now mentioned in each of the figure legends (yellow highlights).
Authors detect that the strongest transcriptonional perturbation is driven by treatment, irresepective of genotype. The authors should evaluate if there are basal gene expression differences between WT and Lrr2-G2019S microglia, or at least state that there are not, if that’s the case.
Also, the authors cuold integrate this dataset with the LPS treated microglia study.
Response: We thank the reviewer for wanting to note down any basal level changes between WT and G2019SLRRK2 mice. We have now included the analysis in the results section 3.4 (Yellow highlight line nos :271-278) and as a Venn diagram visualisation in new Supplementary Figure S2.
We have now compared and contrasted our study with the previous transcriptomic study using LPS where appropriate. The edits are made and highlighted in yellow (Line Nos 302-310).
THe interpretation of the chagnes in lysosomal genes is not clear. zymosan activate microglia, but downregulates Ctsb expression. Authors should discuss how this data agrees or disagrees with previous reports showing that increased expression of Ctsb is associated with increased PD risk.
Response: We thank the reviewer for pointing out this out and we have now added specific discussions around GPNMB and CTSB genes. Highlighted in yellow. (Lines 393-405).
Overall, rationale of the study is not clear. The evaluation of the consequences of zymosan-induced transcriptomic alterations in WT microglia and Lrrk2-G2019S microglia lacks biological significance.
Response: we have now added further clarifications of using zymosan for our experiments. Yellow highlights in the introduction (line 67-71) and also in the discussion (line 311-330). Extra references have also been added.
Reviewer 2 Report
Comments and Suggestions for Authors
The manuscript by Nazish et al. has performed transcriptomic analysis of microglial cell cultures following treatments to stimulate inflammatory responses (zymosan), inhibit LRRK2 kinase (MLi-2) or both in the presence of LRRK2 WT or LRRK2 G2019S. The analysis reveals differential gene epxression induced by zymosan primarily and by the genotype secondarily and GO analysis reveals several affected processes in each tested condition. This work is relevant and interesting, however some improvements are needed such as clarifying experimental details, improving the readability of the figures and most essentially fully overhauling the discussion text.
Given that the biggest effects are obtained using Zymosan treatment, it seems important that some more details on this agent be included in the paper. Zymosan is presented in the introduction as a TLR2 agonist, however little is mentioned about potential other receptors impacted nor about the downstream effects of zymosan such as impacts on phagocytosis, intracellular signalling and/or membrane trafficking.
Related to this, the choice of treatment regimens should be justified, in particular the concentration and time of treatment for both Zymosan and MLi-2. It should also be noted that the concentrations chosen are very high for both and the potential for off target effects should be discussed or at least mentioned.
Figure 1 : It should be clarified whether phospho-Ser935-LRRK2 and total LRRK2 were quantified on the same blots or not. If quantified from the same blot, it should be mentioned if this is via separate secondary antibody detections at different wavelengths or after a membrane stripping step.
Figure 1 : the legend also mentions that the data represent 2 independent expriment each performed in triplicate. It should be clarified whether or not the individual replicates, within each run in triplicate, are biologically independent replicates, resulting in a sample size of N=6. Alternatively, if the triplicates are technical replicates that are not biologically independent, the authors would need to mention a sample size of 2 and adapt the statistical analysis accordingly.
Figure 1, is there an effect observed of MLi-2 alone? It looks like this condition may lead to significant dephosphorylation of LRRK2 at 4h but not at 24h, however no indications are given. Based on what is given in Materials and Methods, an ANOVA with post-hoc Tukey’s test would have been performed. It should be specified in the legend which specific comparisons were made
For the RNAseq experiments, it is not specified what the treatment conditions are aside from the agent used. What are the concentrations and treatment times ?
Figures 3-4-5-5 have very low readability, primarily due to the very small text sizes. Even when zooming the text is very fuzzy and barely readable. A solution should be found to increase font sizes, if needed by breaking down the figures into several separate figures.
Figures should be shown in the order that they are mentioned in the text. In this regard, as the text mentions Figure 3E before Figure 3A, it is necessary to reorganize Figure 3 or reorganize the manuscript text to follow the order of figure referencing in the text.
The main concern with the manuscript is the discussion text which is essentially limited to reiterating the main experimental findings without providing any actual insight. The discussion should be entirely overhauled to provide more in depth insights into the experimental findings and more thorough linking of the findings to previous reports. Just some ideas could be :
- The authors mention that previous transcriptomic studies were done using LPS and theirs is the first to test zymosan. Why did the authors not discuss similarities and differences between their findings with these previous reports with LPS ?
- Several processes are not discussed in relation to previous literature. Is what is identified entirely new conceptually for LRRK2 ? or are there previous studies that have linked LRRK2 to the identified processes ? It may be that it is a mix, however the authors should integrate this and reach a conclusion.
- The top20 analysis is not justified. Why use this cut-off instead of something else such as top10, top50 or other ?
Minor comments
- Abstract line 34 : a space is missing at the end of the sentence
- In text citations : a space should be inserted between the text and the citations bracket.
- Text line 85 : ‘et al’ should be ‘et al.’
- Molecular weight markers are missing in the blots in figure 1.
- ‘et al’ should be ‘et al.’ and be in italics
- What do the authors mean that S935 phosphorylation of LRRK2 is a proxy for LRRK2 activation? Phosphorylation at this site is not clearly linked to the activation of a specific LRRK2 function. If looking at kinase activation of LRRK2, the phosphorylation at S935 is not correlated at all and if anything is rather correlated to reduced LRRK2 kinase activity.
- What are the units on the y-axis of the graphs in Figure 1 ? These should be specified.
Author Response
The manuscript by Nazish et al. has performed transcriptomic analysis of microglial cell cultures following treatments to stimulate inflammatory responses (zymosan), inhibit LRRK2 kinase (MLi-2) or both in the presence of LRRK2 WT or LRRK2 G2019S. The analysis reveals differential gene epxression induced by zymosan primarily and by the genotype secondarily and GO analysis reveals several affected processes in each tested condition. This work is relevant and interesting, however some improvements are needed such as clarifying experimental details, improving the readability of the figures and most essentially fully overhauling the discussion text.
Given that the biggest effects are obtained using Zymosan treatment, it seems important that some more details on this agent be included in the paper. Zymosan is presented in the introduction as a TLR2 agonist, however little is mentioned about potential other receptors impacted nor about the downstream effects of zymosan such as impacts on phagocytosis, intracellular signalling and/or membrane trafficking.
Response: We thank the reviewer for this comment and accordingly have added extra details both in the introduction (lines 67-71) and also in the discussion (lines: 311-330). Yellow highlights.
Related to this, the choice of treatment regimens should be justified, in particular the concentration and time of treatment for both Zymosan and MLi-2. It should also be noted that the concentrations chosen are very high for both and the potential for off target effects should be discussed or at least mentioned.
Response: Previous studies have used 200mg/ml concentration of zymosan in immune cells such as in BMDMs Dzamko et al 2012 (DOI: 10.1371/journal.pone.0060086). For MLi-2, previous studies have used 1mM concentration of MLi-2 based on the paper (in vivo) by Kluss et al 2018 REF: 64; (DOI: 10.1186/s13024-021-00441-8). The experiments were performed at NIH under the guidance and advice of Dr Adamantios Mamais and Dr Mark Cookson. Please also note a dose of 0.5mm MLi2 (in vitro) has been used in the report by Tasegian et al 2021; https://doi.org/10.1042/BCJ20210375. Therefore, we feel we are justified in using the chosen doses for zymosan and MLi2 in this study.
Figure 1 : It should be clarified whether phospho-Ser935-LRRK2 and total LRRK2 were quantified on the same blots or not. If quantified from the same blot, it should be mentioned if this is via separate secondary antibody detections at different wavelengths or after a membrane stripping step.
Response: Membranes to be incubated with LRRK2 and Ser935 were run on separate blots with their respective β-actin. This detail is now added to the paper, (see results section; Line no’s 121-123) (highlighted yellow).
Figure 1: the legend also mentions that the data represent 2 independent expriment each performed in triplicate. It should be clarified whether or not the individual replicates, within each run in triplicate, are biologically independent replicates, resulting in a sample size of N=6. Alternatively, if the triplicates are technical replicates that are not biologically independent, the authors would need to mention a sample size of 2 and adapt the statistical analysis accordingly.
Response: We can clarify that we have performed 2 independent experiments with 3 or 2 biological replicates as detailed in the legends of each figure and therefore the statistics and the representative graphs described is justified. These are now mentioned in each of the figure legends (yellow highlights).
Figure 1, is there an effect observed of MLi-2 alone? It looks like this condition may lead to significant dephosphorylation of LRRK2 at 4h but not at 24h, however no indications are given. Based on what is given in Materials and Methods, an ANOVA with post-hoc Tukey’s test would have been performed. It should be specified in the legend which specific comparisons were made.
Response: We thank the reviewer for this query and we can clarify that there is no statistically significant dephosphorylation of LRRK2 at 4h with MLi-2. Although it seems there may be but the p value of the control vs MLi-2 is p = 0.7137. This is also due to taking into account the results from n=2 experiment.
Statistical comparison was done using One way Anova with Tukey’s post-hoc test. This is now highlighted in the legend.
For the RNAseq experiments, it is not specified what the treatment conditions are aside from the agent used. What are the concentrations and treatment times ?
Response: We have now clarified the treatment timesfor RNA Seq experiments in the methods (line nos 112-114) and also in the legend. (highlighted in yellow). It is an n of two independent experiments with two biological replicates.
Figures 3-4-5-5 have very low readability, primarily due to the very small text sizes. Even when zooming the text is very fuzzy and barely readable. A solution should be found to increase font sizes, if needed by breaking down the figures into several separate figures.
Response: We thank the reviewer for pointing the readability aspect of the figures ad have therefore submitted higher resolution images. Also more specifically we have separated Fig 3 (original ) into 3 separate figs : Fig3,4,5.
Figures should be shown in the order that they are mentioned in the text. In this regard, as the text mentions Figure 3E before Figure 3A, it is necessary to reorganize Figure 3 or reorganize the manuscript text to follow the order of figure referencing in the text.
Response: We thank the reviewer to point out our error and we have addressed this issue accordingly. The Fig nos are now in sequence.
The main concern with the manuscript is the discussion text which is essentially limited to reiterating the main experimental findings without providing any actual insight. The discussion should be entirely overhauled to provide more in-depth insights into the experimental findings and more thorough linking of the findings to previous reports. Just some ideas could be.
Response: We thank the reviewer for pointing to our shortcomings on the discussion and we have now made extensive additions to clarify our results. They are all highlighted in yellow.
- The authors mention that previous transcriptomic studies were done using LPS and theirs is the first to test zymosan. Why did the authors not discuss similarities and differences between their findings with these previous reports with LPS ?
Resposne: We have now compared and contrasted our study with the previous transcriptomic study using LPS where appropriate. The edits are made and highlighted in yellow (Line Nos 302-310).
- Several processes are not discussed in relation to previous literature. Is what is identified entirely new conceptually for LRRK2 ? or are there previous studies that have linked LRRK2 to the identified processes ? It may be that it is a mix, however the authors should integrate this and reach a conclusion.
Response: We have now made several additional comments in the discussion based on LRRK2 functioning in the immune system (yellow highlights).
The top20 analysis is not justified. Why use this cut-off instead of something else such as top10, top50 or other ?
Response: The analysis was done with the entire data but we have presented the top 20 significant gene alterations for ease of representation in the heat maps.
Minor comments
- Abstract line 34 : a space is missing at the end of the sentence
- In text citations : a space should be inserted between the text and the citations bracket.
- Text line 85 : ‘et al’ should be ‘et al.’
- Molecular weight markers are missing in the blots in figure 1.
- ‘et al’ should be ‘et al.’ and be in italics
Response: All of the above minor concerns are now rectified.
- What do the authors mean that S935 phosphorylation of LRRK2 is a proxy for LRRK2 activation? Phosphorylation at this site is not clearly linked to the activation of a specific LRRK2 function. If looking at kinase activation of LRRK2, the phosphorylation at S935 is not correlated at all and if anything is rather correlated to reduced LRRK2 kinase activity.
Response: We agree that the word proxy is not quite valid in this context and have altered it to describe it as an indirect measure of LRRK2 activation. Please see highlighted lines: 156-159.
- What are the units on the y-axis of the graphs in Figure 1 ? These should be specified.
Response: We have now specified the units of the y-axis in modified Fig 1.
Reviewer 3 Report
Comments and Suggestions for Authors
The authors investigated the effects of zymozan, a TLR2 agonist and MLi2 on LRRK2 phosphorylation Level using WB and on the gene expression of primary microglial cells obtained from LRRK2WT and LRRK2p.G2019S knock-in mice using RNAseq. The authors observed an increase expression of the LRRK2 phosphorylation at Ser935 with 2 zymozan treatments at 4 and 24 hours in both genotypes cells, but a decrease expression at this site when cotreated with Zymosan and MLi2 (FIG1). For RNAseq analyses, only the 24hours treatments were examined. The analyses revealed a major effect of the zymosan treatments and to a lesser extent on genotypes of the samples. Zymosan and MLi2 treatments of the microglial cells gave distinct as well as overlapping profiles of differentially expressed genes. Of note, GO term related to inflammatory response was deregulated after Zymosan treatment, while transmembrane transport was upregulated after MLi2 treatment. In addition, cathepsin B and glycoprotein-nmb were down regulated after zymosan treatment.
Comments:
Material and methods: The authors indicate that the statistical test used are mentioned within the legend of the figures, but this is not the case for figure 1. In this figure, while the figures A and B clearly show the effects described by the authors, it is not clear to me how statistical analyses could be conducted in the absence of at least 3 independent experiments (here 2 independent experiments).
It would be good for the reader to mention in the 2.6 section the statistical test used for the RNAseq analyses
Figure 2: Could the authors note in the legend how many independent experiments were performed for the RNAseq?
Figure 3: When printing the document, the quality of this figure is insufficient. It does not allow clearly read the names of the genes (in 3 C, D, E) or conditions (in 3 A, B).
Discussion: in the introduction or discussion, it would be good to explain to the readers not familiar to the LRRK2 field, why choosing as readout the examination of the LRRK2 Ser935 instead of the direct measure of the kinase activity at Ser1292.
Author Response
The authors investigated the effects of zymozan, a TLR2 agonist and MLi2 on LRRK2 phosphorylation Level using WB and on the gene expression of primary microglial cells obtained from LRRK2WT and LRRK2p.G2019S knock-in mice using RNAseq. The authors observed an increase expression of the LRRK2 phosphorylation at Ser935 with 2 zymozan treatments at 4 and 24 hours in both genotypes cells, but a decrease expression at this site when cotreated with Zymosan and MLi2 (FIG1). For RNAseq analyses, only the 24hours treatments were examined. The analyses revealed a major effect of the zymosan treatments and to a lesser extent on genotypes of the samples. Zymosan and MLi2 treatments of the microglial cells gave distinct as well as overlapping profiles of differentially expressed genes. Of note, GO term related to inflammatory response was deregulated after Zymosan treatment, while transmembrane transport was upregulated after MLi2 treatment. In addition, cathepsin B and glycoprotein-nmb were down regulated after zymosan treatment.
Comments:
Material and methods: The authors indicate that the statistical test used are mentioned within the legend of the figures, but this is not the case for figure 1. In this figure, while the figures A and B clearly show the effects described by the authors, it is not clear to me how statistical analyses could be conducted in the absence of at least 3 independent experiments (here 2 independent experiments).
Response: We thank the reviewer for the valuable comments. We can clarify that we have performed 2 independent experiments with 3 or 2 biological replicates as detailed in the legends (modified) of each figure and therefore the statistics and the representative graphs described is justified. These are now mentioned in each of the figure legends (yellow highlights).
It would be good for the reader to mention in the 2.6 section the statistical test used for the RNAseq analyses
Response: We have now mentioned the statistics used for RNA SEQ to normalise the data on in the methods section (line nos 133-135). Highlighted in yellow.
Figure 2: Could the authors note in the legend how many independent experiments were performed for the RNAseq?
Response: We conducted RNA Seq in two independent experiments with two biological replicates. These are highlighted in yellow in the methods section and also in the legends where appropriate.
Figure 3: When printing the document, the quality of this figure is insufficient. It does not allow clearly read the names of the genes (in 3 C, D, E) or conditions (in 3 A, B).
Response: We thank the reviewer for this issue and we have now provided better resolution images.
Discussion: in the introduction or discussion, it would be good to explain to the readers not familiar to the LRRK2 field, why choosing as readout the examination of the LRRK2 Ser935 instead of the direct measure of the kinase activity at Ser1292.
Response: Many thanks for pointing this out. We have now mentioned this in the introduction and in the discussion section regarding Pser935 (Line nos 67-71) and Pser 1292( 426-429). (Yellow highlighted)
Round 2
Reviewer 1 Report
Comments and Suggestions for Authors
I think the paper can be accepted, I would like to suggest that the authors should explain a little bit more, and discuss, the results of the comparison of WT vs G2019S microglia at basal state. The analysis is ok, but the authors do not explain the findings.
Author Response
- I think the paper can be accepted. I would like to suggest that the authors should explain a little bit more, and discuss, the results of the comparison of WT vs G2019S microglia at basal state. The analysis is ok, but the authors do not explain the findings.
Response: We thank the reviewer for this comment and we have now provided a table of genes that were significantly altered in G2019S but not WT mice (see revised Table S1). The premise of our work was to understand the gene expression changes in G2019S vs WT LRRK2 mice with zymosan and MLi-2 treatments and we have discussed this in detail in this manuscript. Further analysis on basal level gene expression is beyond the scope of the current paper and will be tackled in future investigations.
Reviewer 2 Report
Comments and Suggestions for Authors
This version of the manuscript has dealt adequately with most of the concerns raised. Nevertheless, some points that were adressed in the responses to the reviewers have not been integrated in the text and figures and should be:
- the authors have provided a justification of their selection of the treatment regimens (doses and treatment times) for Zymosan and MLi-2 in the responses to this reviewer, however these details need to be provided in the manuscript.
- concerning the use of LRRK2 phosphorylation at S935 as a marker, the authors state that S935-LRRK2 phosphorylation is 'responsive to inhibition of LRRK2 kinase activity and is considered a pharmacodynamic marker'. This is not true given that kinase dead mutants of LRRK2 such as K1906M show no changes in phosphorylation at this site compare to WT-LRRK2. Rather, the authors should specify that this marker is responsive to pharmacological inhibition of LRRK2 kinase activity. The authors also state 'We therefore used S935 phosphorylation as an indirect measure of LRRK2 activation'. Also this is not true as a LRRK2 hyperactivation is observed in disease that correlates to reduced S935 phosphorylation. While the exact significance of this marker is still under study, it is nevertheless a marker indicating shifts in LRRK2 signaling, hence 'activation' wold best be replaced by another word better indicative of the true nature of the pS935-LRRK2 marker.
- the new figures 3-4-5 have not been inserted in the manuscript. These need to be provided in order to reach a final recommendation on whether the manuscript is acceptable or not.
Author Response
This version of the manuscript has dealt adequately with most of the concerns raised. Nevertheless, some points that were addressed in the responses to the reviewers have not been integrated in the text and figures and should be:
- The authors have provided a justification of their selection of the treatment regimens (doses and treatment times) for Zymosan and MLi-2 in the responses to this reviewer, however these details need to be provided in the manuscript.
Response: The dosing details with references are now added in the revised manuscript in section 2.3 (highlighted in green).
- Concerning the use of LRRK2 phosphorylation at S935 as a marker, the authors state that S935-LRRK2 phosphorylation is 'responsive to inhibition of LRRK2 kinase activity and is considered a pharmacodynamic marker'. This is not true given that kinase dead mutants of LRRK2 such as K1906M show no changes in phosphorylation at this site compare to WT-LRRK2. Rather, the authors should specify that this marker is responsive to pharmacological inhibition of LRRK2 kinase activity. The authors also state 'We therefore used S935 phosphorylation as an indirect measure of LRRK2 activation'. Also this is not true as a LRRK2 hyperactivation is observed in disease that correlates to reduced S935 phosphorylation. While the exact significance of this marker is still under study, it is nevertheless a marker indicating shifts in LRRK2 signalling, hence 'activation' would best be replaced by another word better indicative of the true nature of the pS935-LRRK2 marker.
Response: We thank the reviewer for this valuable comment and have made alterations to the text (highlighted in green) in section 3.1.
- The new figures 3-4-5 have not been inserted in the manuscript. These need to be provided in order to reach a final recommendation on whether the manuscript is acceptable or not.
Response: The new figures are now incorporated into the manuscript.
Reviewer 3 Report
Comments and Suggestions for Authors
Thank you for this new version, in which all the comments have been addressed
Since the authors provided additional information (supplementary figure S2) in the paragraph 3.2 it will be good to also insert the information concerning Fold-Change and q-values in the supplementary table S1. In that way it will also highlight subtle differences only due to the genotype. In the same view, GO enrichments analyses might be tested on these basal conditions (up and down or combining up and down regulated genes). If significant these data could also be provided in order to define if some GOs in the LRRK2 G2019S are common to the treatments conditions or at least mention if some deregulated genes are shared with the treatment’s conditions and to comment on this.
Adding the number of mice used per each independent primary microglial culture per genotype will be good in the material and methods since it is not described in the materiel and methods or in the reference 19 (better to cite Russo I J neuroinflammation 2015, doi 10.1186/s12974-015-0449-7, cited in reference 19). This would help to understand the whole procedure and to which extend the biological samples are independent.
Author Response
Thank you for this new version, in which all the comments have been addressed
- Since the authors provided additional information (supplementary figure S2) in the paragraph 3.2 it will be good to also insert the information concerning Fold-Change and q-values in the supplementary table S1. In that way it will also highlight subtle differences only due to the genotype. In the same view, GO enrichments analyses might be tested on these basal conditions (up and down or combining up and down regulated genes). If significant these data could also be provided in order to define if some GOs in the LRRK2 G2019S are common to the treatments conditions or at least mention if some deregulated genes are shared with the treatment’s conditions and to comment on this.
Response: We thank the reviewer for this comment and have now added a list of genes that were significantly changed in G2019S compared to WT (Table S1 updated). The premise of our work was to understand the gene expression changes in G2019S vs WT LRRK2 mice with zymosan and MLi-2 treatments and we have discussed this in detail in this manuscript. Further analysis on basal level gene expression is beyond the scope of the current paper and will be tackled in future investigations.
- In the same view, GO enrichments analyses might be tested on these basal conditions (up and down or combining up and down regulated genes). If significant these data could also be provided in order to define if some GOs in the LRRK2 G2019S are common to the treatments conditions or at least mention if some deregulated genes are shared with the treatment’s conditions and to comment on this.
Response: We thank the reviewer for this comment. Although we understand the need of investigating basal conditions but this will require thorough dissection of gene expression changes and GO enrichment analyses in G2019S vs WT LRRK2 mice which we plan to study in future projects.
- Adding the number of mice used per each independent primary microglial culture per genotype will be good in the material and methods since it is not described in the materiel and methods or in the reference 19 (better to cite Russo I J neuroinflammation 2015, doi 10.1186/s12974-015-0449-7, cited in reference 19). This would help to understand the whole procedure and to which extend the biological samples are independent.
Response: We have now included the number of pups used per experiment in the methods section. However please note that the cell cultures were performed from 3 different flasks to get our independent biological replicates. For obvious sensitive issues we do not want to divulge the exact numbers of pups used in our open access paper. However, if anyone has any queries I am happy to answer them as corresponding author. We have now added the Russo I 2015 paper in the manuscript (Ref 19).